# Don't Encrypt the Data, Just Approximate the Model/
# Towards Secure Transaction and Fair Pricing of Training Data

## Abstract

As machine learning becomes ubiquitous, deployed systems need to be as accurate as they can. As a result, machine learning service providers have a surging need for useful, additional training data that benefits training, without giving up all the details about the trained program. At the same time, data owners would like to trade their data for its value, without having to first give away the data itself before receiving compensation. It is difficult for data providers and model providers to agree on a fair price without first revealing the data or the trained model to the other side. Escrow systems only complicate this further, adding an additional layer of trust required of *both* parties. Currently, data owners and model owners don't have a fair pricing system that eliminates the need to trust a third party and training the model on the data, which 1) takes a long time to complete, 2) does not guarantee that useful data is paid valuably and that useless data isn't, without trusting in the third party with both the model and the data. Existing improvements to secure the transaction focus heavily on encrypting or approximating the data, such as training on encrypted data, and variants of federated learning. As powerful as the methods appear to be, we show them to be impractical in our use case with real world assumptions for preserving privacy for the data owners when facing *black-box* models. Thus, a fair pricing scheme that does not rely on secure data encryption and obfuscation is needed before the exchange of data. This paper proposes a novel method for fair pricing using data-model efficacy techniques such as influence functions, model extraction, and model compression methods, thus enabling secure data transactions. We successfully show that without running the data through the model, one can approximate the value of the data; that is, if the data turns out redundant, the pricing is minimal, and if the data leads to proper improvement, its value is properly assessed, without placing strong assumptions on the nature of the model. Future work will be focused on establishing a system with stronger transactional security against adversarial attacks that will reveal details about the model or the data to the other party.

## 1 Problem Setup

Driven by the application of facilitating data exchange with clear consent, we focus on the most needed form of data transaction: a potential improvement to an already powerful model. A tiny model without sensible accuracy is not proven to work and is likely not deployed for important tasks. Models trained only a small scale of data are furthermore very sensitive to new data of any form, so the question of whether a new dataset brings forth improvement is easy to answer (yes!). A data transaction is the hardest and most meaningful in the setting of a robust model potentially improving with the addition of a relatively small set of data that makes the training set better mimic that of the real world i.e. deployed cases.

Table 1: Privacy, practicality, and fairness comparison of naive exchanges

| Approach | Data leakage | Model leakage | Practicality | Fairness |
|---|---|---|---|---|
| Data Owner gives up data | High | Low | High | Low |
| Model Owner gives up model | Low | High | High | Low |

## 1.1 LIMITATIONS OF THE NAIVE APPROACH

This subsection shows that the naive approach of exchanging data upfront or paying blind upfront are both undesirable, as summarized in Table 1.

In industry, because only tech giants with sufficient existing centrality gets to aggregate the most data, a convenient naive practice involves the data providers giving the data up in advance.

1. An individual user using storage, social, or other valuable features may give up their data within the services for convenience. It can be argued that they are trading their data in for improved services they receive.

2. A small company that provides data to be evaluated, while trusting the auditing within the big companies to prevent useful data from not being paid.

3. Academic researchers in fields of linguistics, biology or other fields, upon hearing about the power of machine learning in industry, give their field studies data collected in decades to model owners, who tend to be entities of big corporations, for free, hoping for collaboration and interesting insights produced.

The fairness in the pricing is highly questionable, as the implicit contracts get difficult to verify and reinforce. Furthermore, the data that is evaluated to be useless currently will likely still sit in the company's cluster, so while the company may decline reciprocating gifts such as academic collaboration, while using the data for some other service in the future. It is difficult for data providers to argue for a latent payment, since any data given up is given up. Improvement of services may never get delivered, and a user of a centralized service who has given up their data will have trouble telling if their data exchange was fair at all (even if their evaluation was purely psychological).

A comparable approach is the other way around: a data company prices their dataset, such as phone call clips, in advance, and a model owner, such as a speech recognition service, pays for the data in full, before any evaluation is done. The fairness in the pricing is dubious, as the value of the data is not evaluated against the model.

While this approach has high security guarantee for one of the involved parties, it is clearly inflexible. For one thing, in both cases, the financial incentives do not align: the model owners would rather not pay fairly for the data while the data owner would rather not price their data fairly. This is further complicated if we look into more granular fairness that relates to the data's interaction with the model: can we reason about the pricing of the data and the presumably positive effect it has on the model.

## 1.2 ASSUMPTIONS

For simplicity, we assume the model owner has a relatively robust large model that can use improvements. A hypothesis is made that a particular data provider may be able to improve the model.

For the simplicity of a privacy argument and for the use case of user data, data here is assumed to be totally private before as far as the involved parties are concerned. That is, there is unknown information about the data that may benefit the model.

### 1.2.1 THE TRAINING MODEL AS A BLACK BOX

For a generally deployed model, it can take any form. Designing a transaction strategy for each one can be time-consuming and difficult to reason about. In addition, the model owner may wish to have ensemble models, or change their model architecture over time. For maximum flexibility, we

assume the model to be a black box. That is, the approaches we take refrain from breaking open the model and making assumptions about it that are too narrow.

## 2 OUR APPROACH: PRICING FUNCTION BASED ON MODEL-DATA EFFICACY

Instead of focusing on encrypting the data and restricting its usage, we propose a different approach which approximates the model instead. It uses what we coin as **Model-Data Efficacy** (MDE) techniques that approximate the data's effect on a particular model for any given model. A MDE choice $f$ is made to work on any model $T$, that provides meaning value on whether the given data $D$ will make changes to the model parameters. Take Spearman's correlation for example, the choice for MDE needs to ideally have

$$d \in D : \rho((f \circ T)(D))_i \sim \rho(T(D))_i.$$

This ensures that the data change under MDE for a given model is similarly ranked against other data points, as with Spearman's Correlation, when compared to the ranking done directly on the model.

Pricing on MDE is used to facilitate a one-time data transaction that is fair; that is, useless data is not paid much money, while useful data gets evaluated.

Our technique imposes a pricing function that is applied to model approximation techniques, which is then applied on given data, as summarized in Figure 4.2.

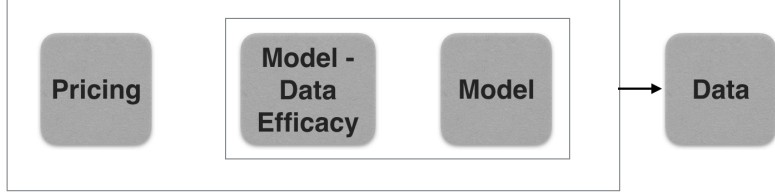

Figure 1: A simplified overview of pricing function.

As summarized in Figure 4.2, a pricing can be put forward in advance before a transaction agreement is put forth; the data owners get paid for the data while the model owners, after paying, will have full flexibility in their usage of the data which has been estimated to be valuable. An improvement is evaluated eventually as the parameter updates that can leader to better performance on a particular test set, which we reduce to just desirable parameter updates.

This choice is a deliberate one. In the next section, we discuss the usability of training on private data for the data transaction use case under our assumptions of a relatively robust but unknown model. The rest of the paper is structured to address other approaches, how practical, secure, private, and fair they are, and give details on our solution.

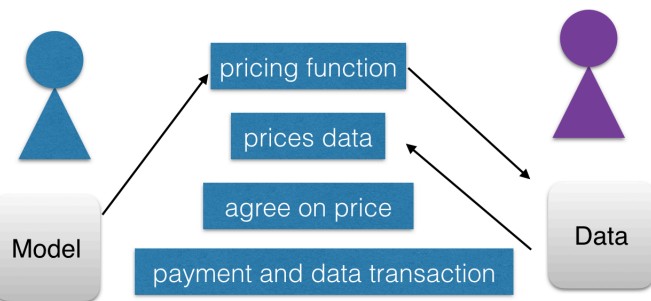

Figure 2: A simplified transaction overview under pricing function. The events happen from top down. Before the transaction, the model is only used to set up a pricing function, while the data is only used to get the resulting price.

## 3 IS THERE USABLE TRAINING DATA PRIVACY?

As machine learning models become more and more complicated, its capability can outweigh the privacy guarantees encryption gives us. Our approach of approximating the model, rather than using techniques to protect the data, is a calculated one: when facing black box models, encrypted training data is not so private. This section addresses the practical implications of private training data in our use case.

### 3.1 DATA THAT IS REVEALED IS GIVEN UP FOREVER

Once a transaction of the raw data is made, the data no longer belongs to the data owner. The model owner is then free to use the data however many times in however many ways unless further restrictions are applied. It is fair to assume that from a security and privacy perspective, any data that is given up is given up forever. Under that assumption, a fair pricing scheme is needed to make sure a pricing happens *before* a transaction, thus giving data owners the opportunity to throughly trade the entirety of the data.

### 3.2 TRANSACTING ONCE

If available, the test set should only be used once to maintain statistical validity, a test set is ideally only used once to prevent model improvements that are just overfitting. Similarly, our data evaluation framework affords a one-time evaluation between every pair of data and model, since repeated testing would leak a lot of contextual information. The proposed method, for instance, assumes the pricing and transaction to be both one-time activities. See the next sections for details.

Note that even though giving the model owners the entirety of the data may lead to overfitting, as Numerai had tried to solve (Richard Craid, 2017), such concern is at a loss of the model owner's, and therefore not a concern of our paper, which aims to align the economic incentives of model owners and data owners.

### 3.3 FACING COMPLICATED MODELS, ENCRYPTING DATA IS A LOST CAUSE

The motivation for training on encrypted data includes user data data and the desire to still improve a learned model. These ideals are tradeoffs. Specifically, for notable improvements on the model the data can demonstrate, some information about the data will need to be leaked, regardless of whether the model input was encrypted.

More practically, it is well-known that popular networks on media data can memorize complete noise (but does not reveal data), as shown by Zhang et al. (2016), making it further difficult to maintain the privacy of data for any meaningful model improvement that the model owner needs to observe, because the parameter updates alone reveal much information about the data.

The following is a sketch of the proof from an information theory standpoint with real-world scales:

Suppose we have a high resolution image passed to a visual network. A practical choice for an image would be a high resolution selfie, such as an $1024x1024$ pixel black and white image. The network is assumed to be a smaller variant of AlexNet (Krizhevsky et al., 2012), which has only 1 million parameters and only outputs binary updates. Note that we restrict the model capability to have a conservative estimate.

Suppose every model parameter update is non-trivial, defined as greater than $\epsilon$ for some $\epsilon$ where $0 < \epsilon \ll 1$. For a given input on the scale of 1MB, once encrypted and trained against an unknown model of 1 million parameters, it outputs binary parameter updates; that is, result $r_i = 1$ for updating $\theta_i$ and $r_i = 0$ for not updating $\theta_i$.

As we know, the model parameters can encapsulate 1 million bits of information, so it has the capacity to model such an input by pixel. This means that there is no good information theoretical guarantee for encrypted data on a completely unknown model of arbitrary scale, especially in the use cases today.

In fact, even for unencrypted data, the model owner should be incentivized to not reveal the details of its model and should instead use a proxy for the model in evaluating the efficacy of data.

### 3.4 A CASE FOR USING RAW TRAINING DATA IN MACHINE LEARNING

In practice, unrestricted raw training data is useful for model improvement. In addition, extremely restricted visibility into data can be dangerous, as in the case of training on encrypted data. A practical solution thus calls for more flexibility in data usage for the model owners.

The model owners' incentives are against replacing the entire data with much less information, such as with the result of feature extractions; because encrypted data is undesirable, the model owners will be less likely to employ it. Firstly, the engineering work involved each time such a filter of information is made is expensive. This adds to the cost of training on encrypted data, further dis-incentivizing model owners from sticking to the encryption scheme that requires regular updates. More broadly, the model owner would like to sometimes change the architecture of the model to improve it; not having the underlying training data stored makes such process hard.. If a feature extractor add one more entry, all the data would need to be extracted again. Similarly, if an image classifier's model architecture changes, all the data would need to be collected and purchased again.

Even with the same model, extremely restricted data is inflexible for the model owners in improving their models. As deep learning matures as a technology, modern development techniques for a safe and robust model requires the model owners to have intimate access to data. The model developers usually require a lot of time, tweaking, test, and in general more iterations of runs with the training data in order to form a better model in the development phase. This development phase is not separable in practice: visibility into why their model works and does not work often rely on examining the data. In the case of media input, the raw data is often more human readable than their representations.

In addition, not having the actual data at the disposal of model owners is undesirable and potentially unsafe. For example, adversarial training data cause misclassification (Goodfellow et al., 2014); autonomous driving cars that mis-classify a stop sign can potentially cause human deaths. To remedy such risks, practical model testing and debugging methods often rely on visibility into data(Koh & Liang, 2017); for instance, adversarial images that cause mis-classification need to be examined

Table 2: Privacy, practicality, and fairness comparison of advanced methods.Encrypting the model is impractical in the case of large models that are deployed, thus the high fairness hypothesis is not applicable. Encrypting the data has data leakage that is specific to the model, which is marked as medium.

| Approach | Data leakage | Model leakage | Practicality | Fairness |
|---|---|---|---|---|
| Escrow, smart contract | Medium | Medium | Low | High |
| Encrypting the model | High | Low | Low | N/A |
| Encrypting the data | Medium | Low | Low | Medium |

Table 3: Privacy, practicality, and fairness comparison of federated learning (with encrypted model and private, verifiable aggregators) and our proposed pricing with model-data efficacy
.

| Approach | Data leakage | Model leakage | Practicality | Fairness |
|---|---|---|---|---|
| Federated Learning | Low | Low | Low | High |
| Model-Data Efficacy | Low | Low | Medium | High |

thoroughly more than just looked at (Dalvi et al., 2004). Each of the testing technique require yet a new operation on the data; if we want the data to remain encrypted, each of these testing frameworks would need to be hand-crafted to accommodate.

## 4 COMPARING WITH RELATED WORKS IN PRIVACY-PRESERVING TRAINING

Fully homomorphic encryption, verifiable contracts and proofs, and federated learning techniques address the issue of privacy-preserving training. We talk about their privacy, practicality, and fairness tradeoffs in this section, as summarized in Table 2. In particular, we discuss the beneficial intuitions in federated learning, as summarized in Table 3.

### 4.1 FULLY HOMOMORPHIC ENCRYPTION

Homomorphic encryption is appealing in the privacy and security community for its power to complete verifiably correct computation without revealing the original inputs. Naively it can be applied to either the data or the model, yet the computational cost is prohibitively expensive; as Yonetani et al. (2017) show, to learn a classifier for recognizing 101 objects using 2048-dimensional deep features, a verifiable summing operation encrypted with a 1024-bit key takes about 3 ms for a single weight using a modern CPU on Mac OS X, thus expecting to take up more than 16 hours to encrypt all classifiers even in a simple case (Yonetani et al., 2017). Encrypting the model is thus not practical.

Encrypting the data for machine learning, while still slow, has led to many observations in the differential privacy community. For instance, some models have shown to have secure *and* efficient solutions, such as ridge regressionGiacomelli et al. (2017). Here, we focus on those with strong security guarantees that are in practical use.

### 4.2 BLOCKCHAIN SMART CONTRACT SOLUTION

The advances of bitcoin-related research touch on aligning financial incentives with machine learning algorithms' effectiveness using blockchain. Numerai (Richard Craid, 2017) addresses economic incentive against overfitting, while Openmined (et al., 2017) encourages a collaboration of open protocols and development that incorporate all of federated learning, homomorphic encryption, and blockchain smart contracts, which are all covered here.

Advances in using blockchain smart contracts often have use cases that are diverge from our goal, making it suboptimal for security, flexibility, and pricing fairness. Security-wise, the data detail may

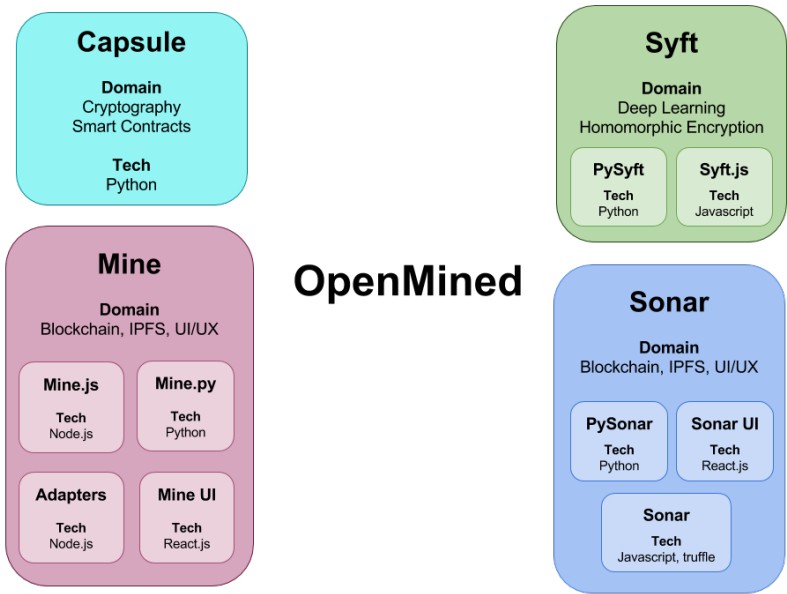

Figure 3: OpenMined architecture, which aims to democratize data by making it easier to exchange user data. It consists of federated learning server Sonar running on the blockchain, Capsule for handling encrypted key exchange, Mine with user data, and Syft, which contains neural networks that can be trained in an encrypted state (et al., 2017)
.

remain secret, yet it can be inferred from the model improvements. To mitigate that, a smart aggregator is designed to rely on the other users' data to mix up the updates, so that such information regarding data detail is differentially private. However, making strong assumptions about the distribution of other data not in our control is impractical, as we deal with a single transaction between a model owner and a data owner. In the case of training data transaction, we want the data to be preferably visible to maximize their flexibility. Absolute security of the data is also less relevant in the pricing case, as we hope to guard the data against its use case; that is, we aim to prevent its specific model improvements to known while still demonstrating that the model is improved by the data.

### 4.2.1 FEDERATED LEARNING

Federated learning has seen great advances, without sacrificing much accuracy, while preserving privacy (et al., 2017) (McMahan et al., 2016). It potentially can be used to allow model parameter updates to be estimated before the central big model gets that information. That is, a transaction can happen between the gradient updates are passed back, and a pricing can be done based on the gradient, achieving the desired privacy guarantee for data owners. This method is an improvement on all the previous methods mentioned when it comes to privacy for data owners who can benefit a lot from the exchange.

However, it is relatively unsafe for model owners, as a miniature version of the model is deployed per client, lacking true ownership, making the attack vector for knowledge about the model very large. In addition, it still requires the effect of the exact data on the exact model to be known before transaction. It further calls for a secure aggregating solution, though existing ones proposed appears to work in experimental results in the case of visual learning (Yonetani et al., 2017).

Our approach can be seen as an improvement on federated learning scheme in that the data owners' privacy is prioritized over the model owner's model details.

Federated learning often involves neural networks that can be trained in an encrypted state, in order to retain the information regarding the model, preventing it from being stolen. Thus they can be very slow. To optimize for them, the distributed model is hand-crafted, which is time consuming.

This encryption guarantee is necessary for federated learning, since the architecture requires the model to be distributed per user, making the attack vector for model theft very large. Under such constraint, speeding up a neural network that is trainable in an absolutely encrypted state has little black box solution that is plug-and-play.

Our method can be seen as an improvement on the federated learning scheme, utilizing the intuition that both the model and the data can be protected to achieve fairness.

### 4.3 ADDING PRICING TO SECURE TRANSACTION

To secure the data, a pricing scheme is needed to reveal as little data detail to the model owners as possible before a transaction happens.

## 5 SYSTEM SETUP AND PRACTICAL REMARKS

The eventual system the data is tested on would output a scalar price for given data. It consists of a pricing component, a model-data efficacy middleware, and the model itself.

The middleware is prepared by the model owners, who have control over the details of their model. A safer mediation is abstracting the middleware into two parts: a model approximation and an efficacy calculation layer.

To model owner, the construction of MDE is under their control; for simplicity, we use a binary score. Then it suffices to have the data causing positive output be paid a uniform price while the data causing zero output to not be transacted, as they are likely useless.

We are formulating the efficacy of data with respect to the model $T$. Such efficacy, denoted as $\rho$ can be seen as a function that loosely maps data to a scalar. Since we care less about the cases when the data does not have value to us, including when the data is a duplication, of high similarly, leads to no parameter change, no model improvement, or even regression, we will only consider the non-negative cases.

### 5.1 EXAMPLES OF MODEL-DATA EFFICACY MIDDLEWARE

For the weak results we wish to obtain: useful versus not very useful, many existing methods can be utilized. Interpretability solutions aim to alleviate the notoriety of reasonability of neural networks. Because the interpretation steps are generally faster or less resource-draining than running the data through the original model, some of these techniques are suitable for our purposes. Leaning on these, we can set up a general framework $f$ where the model $T$'s effect under data $D$ is approximated:

#### 5.1.1 INFLUENCE FUNCTIONS

Influence functions are very flexible: it is shown that the approximation of the effect of the model works well on both seen and unseen data, making it suitable for our purposes (Bastani et al., 2017). The model access influence function needs are gradients and Hessian-vector products, effectively treating the model itself as a blackbox. Valuable results are obtained in evaluating experimental data efficacy even on non-convex and non-differentiable models. The versatility of influence functions is a good direct MDE measure to be used for pricing.

#### 5.1.2 MODEL EXTRACTIONS

A similar approach that aims at interpretability of neural network models (often difficult to reason about) is model extraction (Bastani et al., 2017), which effectiveness approximates model as a decision tree.

Model Extraction is also experimented to have better data efficacy approximation than just training a decision tree from the same set of data. This suggests that the extract model preserves the behavior

of the original model well; that is, data that leads to improvements in model give the same inference results in the extracted model as the original model would, and useless data that does not shift original model will not shift the extracted model. Leveraging that comes a fast approach of evaluating data efficacy, thus providing fair pricing approximations.

### 5.1.3 EXTENSIONS

Besides interpretability techniques through data, a more generic class of techniques that make the model smaller and faster (and less energy-consuming) can be utilized for pricing purposes, because they don't reveal the full model while still retaining its behavior.

There are many techniques on model compression (Han et al., 2015) (Hinton et al., 2015), and our selection for pricing models requires the underlying model compression algorithms to take on relatively black-box approaches to maintain generality and secrecy.

Model compression techniques are very powerful in reducing the resource footprint of large models while retaining the overall accuracy.

Another extension to approximate data efficacy on model includes training an ensemble model that assumes sparsity assumption on models. Because this is closer to just replacing the existing model, rather than keeping it as is while shrinking it, we will leave it to future work to explore the possibilities.

Some techniques are model-specific, such as model distillation for ensemble models (Hinton et al., 2015).

### 5.2 CHOOSING A PRICING FUNCTION

A pricing model takes a data efficacy estimates and outputs a price. The model owners selects the desirable parameter updates.

### 5.3 ENCRYPTING THE DATA EVALUATION PIPELINE

Because we have a much smaller component involved to evaluate the training data, thus encrypting it becomes a lot more practical than encrypting the whole model.

### 5.4 DATA SECURITY

Because we simplified the communication between data owners and model owners to be of a one-time scaler, the data security issue is largely simplified. Once the price is agreed upon, a verifiable transaction is trivial to implement to ensure that the same data is transacted after the money is paid.

## 6 CONCLUSION

Encrypting the data or approximating the data is a lost cause for the data owner whose privacy is not guaranteed. Since the model owner has greater context on similar data distribution, they can infer much information about the data without actually seeing it. Because data cannot be practically secured without losing its value before being handed over, pricing and the transactional form relevant. In this scheme, no data is given up until money is paid.

The suggested methods for Model-Data Efficacy include influence, which explores the change in parameter with training data, model extractions, which approximate a trained network with a decision tree, and model compression techniques that are learned. They all work to approximate the effect of data to the model owners without showing the exact makeup of the model.

The crux of the usability of the solution lies in whether the approximation technique preserves model details, but combining secure transaction techniques is sufficient to make the approximated pricing model entirely private (beyond its output) without further approximating the effect of these pricing models, thus keeping them as accurate as the previous results in the last section.

Despite the potential accuracy loss, usability is much better. For any transaction reached through model approximation, we still maintain usable privacy guarantee. Securing a pricing function, which is very small, is easy. Enforcing by ways of contract to guarantee that a money-data transaction happens after agreeing on a price is much easier to enforce than contracts that bind within the large model owners? organization, such as trusting a security audit.

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
