# OpenReview forum: "Don't encrypt the data; just approximate the model \ Towards Secure Transaction and Fair Pricing of Training Data"
_ICLR.cc/2018/Conference — Reject_

### Official Review · AnonReviewer3 · 2017-11-26
**A paper, on the topic of model/data provider transactions, that is quite unclear w.r.t. the main idea, the connection to previous work, and evaluation of the proposed method**

**Rating:** 2
**Confidence:** 4

**Review:**

Summary

The paper addresses the issues of fair pricing and secure transactions between model and data providers in the context of machine learning real-world application.

Major

The paper addresses an important issue regarding the real-world application of machine learning, that is, the transactions between data and model provider and the associated aspects of fairness, pricing, privacy, and security.

The originality and significance of the work reported in this paper are difficult to comprehend. This is largely due to the lack of clarity, in general, and the lack of distinction between what is known and what is proposed. I failed to find any clear description of the proposed approach and any evaluation of the main idea.

Most of the discussions in the paper are difficult to follow due to that many of the statements are vague or unclear. There are some examples of this vagueness illustrated under “minor issues”. Together, the many minor issues contribute to a major communication issue, which significantly reduces readability of the paper. A majority of the references included in the reference section lack some or all of the required meta data.

In my view, the paper is out of scope for ICLR. Neither the CFP overview nor the (non-exhaustive) list of relevant topics suggest otherwise. In very general terms, the paper could of course be characterised as dealing with machine learning implementation/platform/application but the issues discussed are more connected to privacy, security, fair transactions, and pricing.

In summary; although there is no universal rule on how to structure research papers, a more traditional structure (introduction, aim & scope, background, related work, method, results, analysis, conclusions & future work) would most certainly have benefitted the paper through improved clarity and readability. Although some interesting works on adversarial learning, federated learning, and privace-preserving training are cited in the paper, the review and use of these references did not contribute to a better understanding of the topic or the significance of the contribution in this paper. I was unable to find any support in the paper for the strong general result stated in the abstract (“We successfully show that without running the data through the model, one can approximate the value of the data”).

Minor issues (examples)

- “Models trained only a small scale of data” (missing word)
- “to prevent useful data from not being paid” (unclear meaning)
- “while the company may decline reciprocating gifts such as academic collaboration, while using the data for some other service in the future” (unclear meaning)
- “since any data given up is given up ” (unclear meaning)
- “a user of a centralized service who has given up their data will have trouble telling if their data exchange was fair at all (even if their evaluation was purely psychological)” (unclear meaning)
- “For a generally deployed model, it can take any form. Designing a transaction strategy for each one can be time-consuming and difficult to reason about” (unclear meaning)
- “(et al., 2017)” (unknown reference)
- “Osbert Bastani, Carolyn Kim, and Hamsa Bastani. Interpreting blackbox models via model extraction, 2017” (incomplete reference data)
- “Song Han, Huizi Mao, and William J. Dally. Deep compression: Compressing deep neural networks with pruning, trained quantization and huffman coding, 2015.
Geoffrey Hinton, Oriol Vinyals, and Jeff Dean. Distilling the knowledge in a neural network, 2015.
Pang Wei Koh and Percy Liang. Understanding black-box predictions via influence functions, 2017.” (Incomplete reference data)
- “H. Brendan McMahan, Eider Moore, Daniel Ramage, Seth Hampson, and Blaise Agera y Arcas. Communication-efficient learning of deep networks from decentralized data. 2016.” (Incomplete reference data)
- “et al. Richard Craid.” (Incorrect author reference style)
- “Ryo Yonetani, Vishnu Naresh Boddeti, Kris M. Kitani, and Yoichi Sato. Privacy-preserving visual learning using doubly permuted homomorphic encryption, 2017.
Chiyuan Zhang, Samy Bengio, Moritz Hardt, Benjamin Recht, and Oriol Vinyals. Understanding deep learning requires rethinking generalization, 2016.” (Incomplete reference data)

---

### Official Review · AnonReviewer1 · 2017-11-26
**Review of "Don't encrypt the data; just approximate the model \ Towards Secure Transaction and Fair Pricing of Training Data"**

**Rating:** 4
**Confidence:** 5

**Review:**

The paper discusses a setting in which an existing dataset/trained model is augmented/refined by adding additional datapoints. Issues of how to price the new data are discussed in a high level, abstract way, and arguments against retrieving the new data for free or encrypting it are presented.

Overall, the paper is of an expository nature, discussing high-level ideas rather than actually implementing them, and does not  experimentally or theoretically substantiate any of its claims. This makes the technical contribution rather shallow. Interesting questions do arise, such as how to assess the value of new data and how to price datapoints, but these questions are never addressed (neither theoretically nor empirically). Though main points are valid, the paper is also rife with informal statements  and logical jumps, perhaps due to the expository/high-level approach taken in discussing these issues.

Detailed comments:

The (informal) information theoretic argument has a few holes. The claim is roughly that every datapoint (~1Mbyte image) contributes ~1M bits of changes in a model, which can be quite revealing. As a result, there is no benefit from encrypting the datapoint, as the mapping from inputs to changes is insecure (in an information-theoretic sense) in itself. This assumes that every step of stochastic gradient descent (one step per image) is done in the clear; this is not what one would consider secure in cryptography literature.  A secure function evaluation (SFE) would encrypt the data and the computation in an end-to-end fashion; in particular, it would only reveal the final outcome of SGD over all images in the dataset without revealing any intermediate steps. Presuming that the new dataset is large (i.e., having N images), the "information theoretic" limit becomes ~N x 1Mbyte inputs for ~1M function outputs (the finally-trained model). In this sense, this argument that "encryption is hopeless" is somewhat brittle.

Encryption-issues aside, the paper would have been much stronger if it spent more effort in formalizing or evaluating different methods for assessing the value of data. The authors approach this by treating the ML algorithm as a blackbox, and using influence functions (a la Bastani 2017) to assess the impact of different inputs on the finally trained model (again, this is proposed but not implemented/explored/evaluated in any way). This is a design choice, but it is not obvious. There is extensive literature in statistics and machine learning on the areas of experimental design and active learning. Both are active, successful research areas, and both can be provide tools to formally reason about the value of data/labels not yet seen; the paper summarily ignores this literature.


Examples of imprecise/informal statements:

"The fairness in the pricing is highly questionable"
"implicit contracts get difficult to verify"
"The fairness in the pricing is dubious"
"As machine learning models become more and more complicated, its (sic) capability can outweigh the privacy guarantees encryption gives us"
"as an image classifier's model architecture changes, all the data would need to be collected and purchased again"
"Interpretability solutions aim to alleviate the notoriety of reasonability of neural networks"

---

### Official Review · AnonReviewer2 · 2017-11-27
**one of the official anonymous reviews**

**Rating:** 3
**Confidence:** 5

**Review:**

This paper's abstract is reasonably interesting and has importance given the landscape that is developing.  Unfortunately, however, the body of the paper disappoints, as it has no real technical content or contribution.  The paper also needs a spelling, grammar, typesetting, and writing check.

I don't mind the restriction of the setting under study to be adding a small dataset to a model trained on a large dataset, but I don't agree with the way the authors have stated things in the first paragraph of the paper because there are many real-world domains and applications that are necessarily of the small data variety.

In Section 3.3., the authors should either make a true information-theoretic statement or shorten significantly.

---

### Decision · Program_Chairs · 2018-01-29
**ICLR 2018 Conference Acceptance Decision**

**Decision:**

Reject

**Comment:**

The reviewers highlight a lack of technical content and poor writing.
They all agree on rejection.
There was no author rebuttal or pointer to a new version.